# Phonon downconversion to suppress correlated errors in superconducting qubits

V. Iaia[1,3], J. Ku[1,3], A. Ballard[1], C. P. Larson[1], E. Yelton[1], C. H. Liu[2], S. Patel[2], R. McDermott[2] & B. L. T. Plourde[1] ✉

Quantum error correction can preserve quantum information in the presence of local errors, but correlated errors are fatal. For superconducting qubits, high-energy particle impacts from background radioactivity produce energetic phonons that travel throughout the substrate and create excitations above the superconducting ground state, known as quasiparticles, which can poison all qubits on the chip. We use normal metal reservoirs on the chip back side to downconvert phonons to low energies where they can no longer poison qubits. We introduce a pump-probe scheme involving controlled injection of pair-breaking phonons into the qubit chips. We examine quasiparticle poisoning on chips with and without back-side metallization and demonstrate a reduction in the flux of pair-breaking phonons by over a factor of 20. We use a Ramsey interferometer scheme to simultaneously monitor quasiparticle parity on three qubits for each chip and observe a two-order of magnitude reduction in correlated poisoning due to background radiation.

Qubits formed from superconducting integrated circuits are one of the leading systems for implementation of a fault-tolerant quantum computer[1]. For sufficiently high gate fidelity, error correction schemes such as the surface code[2] can mitigate local errors. However, recent work has shown that high-energy particle impacts from low-level radioactivity and cosmic-ray muons will generate nonequilibrium quasiparticles (QPs)[3–5] that can lead to correlated errors across a multiqubit array[6–8]. Such correlated errors cannot be mitigated by current error correction schemes, thus posing a significant challenge to realization of a fault-tolerant quantum computer.

Particle impacts deposit energy of order 100 keV in the device substrate, leading to the generation of large numbers of electron-hole pairs and a cascade of high-energy phonons[6]. These phonons travel throughout the chip and break Cooper pairs with high probability when they scatter off superconducting structures on the device layer, thus generating QPs at arbitrary locations relative to the particle impact site[6,7,9]. Prior work has explored low-gap superconducting structures for phonon downconversion to protect superconducting resonators and detectors with a higher gap energy[10,11]. Another scheme involves placing superconducting detectors on thin suspended membranes[12]. The use of normal metal layers on the back side of

superconducting qubit chips was proposed in ref. 7 to downconvert energetic phonons below the superconducting gap. Because this downconversion process is based on the scattering of phonons with conduction electrons in the metal, the low rate of electron-phonon scattering at low temperatures[13] dictates large volumes of normal metal for efficient phonon downconversion. A calculation in ref. 7 based on the phonon scattering rate in the normal metal on the back side and the pair-breaking rate in the superconducting film on the device layer indicates that achieving a 100-fold improvement in the qubit energy relaxation time $T_1$ in the aftermath of a particle impact requires a 6-μm-thick normal metal layer.

Here we implement this idea using thick electroplated Cu reservoirs to promote downconversion of phonons below the superconducting gap edge. To test this approach in a controlled way, we integrate Josephson junctions around the chip perimeter and controllably bias the junctions above the superconducting gap to generate pair-breaking phonons on demand. With explicit phonon injection, we find that the phonon-downcoversion structures reduce the flux of pair-breaking phonons by more than a factor of 20. We also examine the correlated errors in multiqubit chips with and without Cu reservoirs and find a two order of magnitude reduction in correlated error rate, to

[1]Department of Physics, Syracuse University, Syracuse, NY 13244-1130, USA. [2]Department of Physics, University of Wisconsin-Madison, Madison, WI 53706, USA. [3]These authors contributed equally: V. Iaia, J. Ku. ✉e-mail: bplourde@syr.edu

the point where these errors no longer pose a limit to fault-tolerant operation.

## Results

### Experimental design

The experimental geometry is shown in Fig. 1. We study two nominally identical chips, one with back-side normal metallization (Cu chip) and one without (non-Cu chip). Each chip incorporates an array of charge-sensitive transmon qubits, with Josephson injector junctions arrayed around the perimeter of the chip. Each qubit has a readout resonator that is inductively coupled to a common feedline that can be used for multiplexed readout. We measure both chips in the same low-temperature environment on the same cooldown. For our qubits, we target a somewhat low ratio $E_J/E_c$ of Josephson energy to single-electron charging energy to produce a peak-to-peak charge dispersion between 1–5 MHz. This allows us to monitor QP parity switching for each qubit, which is a sensitive measure of QP poisoning[14–16]. For the experiments presented here, we focus on three of the qubits on each chip: $Q_A$, $Q_B$, $Q_C$ [Fig. 1b]. Details of the qubit parameters and experimental configuration are given in Methods and Supplementary Notes 3 and 4.

The normal metal reservoirs on the Cu chip consist of 10-μm thick islands patterned from Cu films grown by electrodeposition onto the back side of a high-resistivity double-side polished Si wafer following electron-beam evaporation of a Ti/Cu seed layer. The 10-μm thickness was chosen based on the estimate from ref. 7. The islands are defined with a lattice of partial dicing saw cuts through the Cu film into the back side of the wafer, resulting in island areas of $(200\,\mu m)^2$ [Fig. 1c] (see Methods and Supplementary Notes 1 and 2). This is done to suppress damping from coupling to the transmission line mode formed by a continuous metal layer on the back side of the chip and the ground plane on the device layer that would otherwise degrade

qubit coherence[7]. Metallic losses due to capacitive coupling between the qubit and Cu island are projected to limit the qubit quality factor to ~3M for the qubit design considered in ref. 7; the smaller qubit island size for our qubits reduces this coupling and raises this quality factor limit, thus making a negligible impact on $T_1$.

The injector junctions are fabricated at the same time as the qubit junctions with a standard Al-AlO$_x$-Al process. The ground plane, qubit capacitor islands, readout resonators, and injector junction pads are all fabricated from Nb. There is no direct galvanic connection from the injector junction pads to ground, so the QP poisoning proceeds via phonon emission. By biasing the injector junction above $2\Delta_{Al}/e$, where $\Delta_{Al}$ is the superconducting energy gap for Al, we break Cooper pairs and generate local QPs that subsequently recombine, emitting phonons. These phonons then travel through the Si, and, upon encountering Al junction electrodes for a qubit, a phonon will break a Cooper pair with high probability and generate two QPs. A similar injection scheme was used in refs. 17, 18. Although the phonons injected by the tunnel junction will be lower in energy than those generated by a particle impact, the tunnel junction gives us the ability to control the timing, duration, and location of the phonon injection, in contrast to phonons from particle impacts, which occur at random times and locations.

### Enhanced relaxation from phonon injection

In a first series of experiments, we measure the energy relaxation time $T_1$ of all three qubits on each chip following pulsed QP injection. Here we focus on $Q_C$ ($Q_B$) for the non-Cu (Cu) chip, but we observe similar behavior for the other two qubits on each chip. We use a standard inversion recovery measurement to probe $T_1$. To quantify degradation in $T_1$, we plot $\Delta\Gamma_1 = 1/T_1 - 1/T_1^b$, where $T_1^b$ is the baseline relaxation time from an average of several $T_1$ measurements with no injector junction bias (see Methods). The change $\Delta x_{qp}$ in reduced QP density in the qubit junction leads can be calculated from $\Delta\Gamma_1$[19] (see Methods).

We start by applying a 10-μs injection pulse with amplitude $V_b = 1$ mV, well beyond $2\Delta_{Al}/e$, so that we expect significant QP poisoning in the absence of any mitigation. We vary the delay time between the end of the injection pulse and the $X$ pulse at the start of the $T_1$ sequence. For the non-Cu chip, $\Delta\Gamma_1$ increases substantially following the injection pulse and reaches a maximum poisoning level about 30 μs after the end of the injection pulse [Fig. 2a]. This delayed onset of poisoning is consistent with the propagation timescale for the injected phonons to diffuse through the substrate to the qubit junction. In the absence of phonon downconversion structures, phonons travel throughout the substrate following boundary-limited diffusion, where they scatter randomly off the top and bottom surfaces of the Si chip. This leads to an effective diffusivity $D = c_s d$, where $c_s = 6 \times 10^3$ m/s is the speed of sound in Si and $d = 0.525$ mm is the chip thickness. Thus, the timescale for phonons to diffuse from the injector junction to each qubit (at a separation of 4–6 mm from the injector) is of the order of 10 μs. Following this peak, $\Delta\Gamma_1$ recovers towards the unpoisoned baseline level following an exponential decay with a characteristic time of ~60 μs (see Supplementary Note 5). This corresponds to the timescale for phonons to exit the substrate at the chip perimeter where the sample is acoustically anchored to the device enclosure. The chip is attached to the machined Al enclosure using a small amount of low-temperature adhesive (GE varnish) at the corners of the chip.

For the Cu chip, $\Delta\Gamma_1$ is difficult to distinguish from the baseline level for all delays. For a 30-μs delay, $\Delta\Gamma_1$ for the non-Cu chip is over a factor of 35 larger than for the Cu chip. This is our first key result demonstrating the effectiveness of Cu reservoirs in reducing phonon-mediated QP poisoning.

We then explore the variation of $\Delta\Gamma_1$ with $V_b$ for a fixed delay of 30 μs and a 10-μs pulse width. For the non-Cu chip, we observe a significant increase in $\Delta\Gamma_1$ when the pulse amplitude exceeds $2\Delta_{Al}/e \approx 0.4$ mV. For the Cu chip, $\Delta\Gamma_1$ doesn't change significantly at

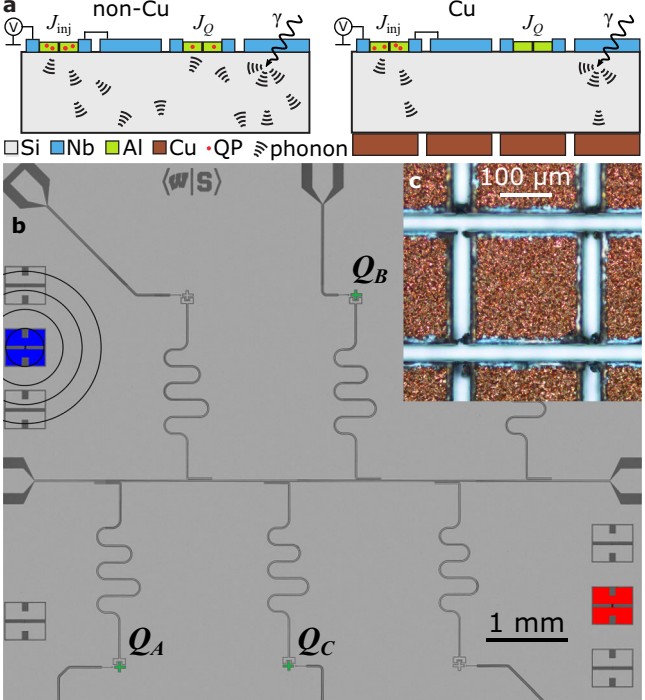

**Fig. 1 | Phonon-mediated QP poisoning and device layout. a** Schematic showing QP/phonon injection, γ impact, phonon propagation in substrate, and pair breaking in qubit junctions with and without Cu. **b** Optical micrograph of device layer. Qubits ($Q_{A,B,C}$) are colored green. Junctions used to inject QPs into the Cu (non-Cu) chip are highlighted in blue (red); concentric rings represent propagating phonons. **c** Optical micrograph of Cu islands on back side of Cu chip.

$2\Delta_{Al}/e$; however, we observe a gradual rise in $\Delta\Gamma_1$ to a peak at a pulse amplitude of 0.56 mV, followed by a reduction to the baseline level for larger pulse amplitudes. We understand the peak in $\Delta\Gamma_1$, which is also visible for the non-Cu chip on top of the overall poisoning curve, to be due to photon-assisted poisoning from absorption of Josephson radiation emitted by the injector junction mediated by a spurious mm-wave resonance in the qubit (see Supplementary Notes 5 and 6). Such antenna effects in qubit structures can lead to resonant absorption of electromagnetic radiation, which can drive high-frequency currents through the qubit junction and generate QPs[20–22]. We would not expect the Cu reservoirs to have any effect on this photon-based QP poisoning mechanism.

## QP parity switching

In a separate series of experiments, we exploit the non-negligible charge dispersion of our qubits to probe the charge parity of the qubit islands as a sensitive probe of QP poisoning. We employ a Ramsey pulse sequence to map QP parity onto qubit 1-state occupation[14,15,23] (see Methods). We first perform the QP parity switching measurement on each of the qubits at a repetition period of 10 ms and then compute

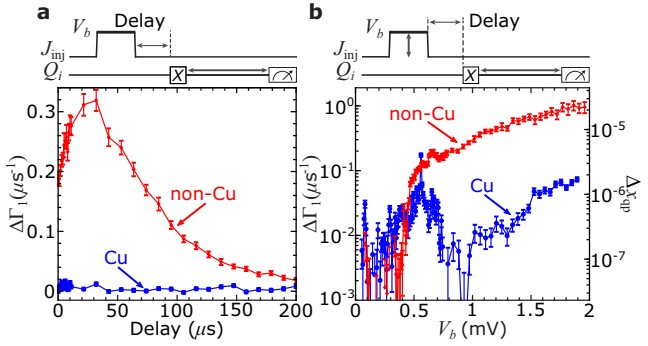

**Fig. 2 | Suppression of $T_1$ from controlled phonon injection. a** $\Delta\Gamma_1$ vs. delay following injection pulse for $Q_C$ on non-Cu (red) chip and $Q_B$ on Cu (blue) chip with $V_b = 1$ mV. **b** $\Delta\Gamma_1$ vs. $V_b$ for non-Cu and Cu chips with 30-μs delay. Error bars computed from 95% confidence intervals from $T_1$ fits (see Supplementary Note 5).

the power spectral density of parity switches. We fit a Lorentzian to the measured spectrum for each chip to extract the characteristic parity switching rate $\Gamma_p$, as in ref. 14. In Fig. 3a, we plot typical spectra from one qubit on each chip. The resulting values for $\Gamma_p$ for both chips are low: $\Gamma_p = 0.360$ s$^{-1}$ (0.023 s$^{-1}$) for the non-Cu (Cu) chip. To the best of our knowledge, $\Gamma_p$ for the non-Cu chip is consistent with the lowest rates for QP poisoning reported in the literature[24,25], while for the Cu chip our measured poisoning rate is an order of magnitude lower.

The low QP poisoning rates on both chips are likely due to a combination of best practices for shielding, filtering, and thermalization (see Supplementary Note 3). In addition, the relatively compact qubit design results in a rather high fundamental antenna resonance frequency, ~270 GHz (see Supplementary Note 6), which is likely above the cutoff of the spectrum of blackbody radiation from higher temperature stages of the cryostat. Following the analysis in ref. 22 applied to the geometry of our qubits, we calculate an effective blackbody temperature of 330 mK (280 mK) for $Q_A$ on the non-Cu (Cu) chip from our measured $\Gamma_p$ values. In addition to photon absorption by the spurious antenna resonance, the residual QP poisoning is likely due to high-energy particle impacts or other radiation sources that generate pair-breaking phonons. We attribute the even lower QP parity switching rate for the Cu chip to absorption of a significant fraction of the phonons generated from these poisoning events by the Cu islands on the back side of the chip, which we will subsequently quantify.

The low QP baseline poisoning rates allow us to directly investigate QP poisoning in the presence of controlled injection of pair-breaking phonons into the chip. Here, we sample QP parity on the qubits with a 100-μs repetition period, while pulsing the injector junction at an amplitude of 1 mV and a fixed rate of 20 Hz. Since this experimental duty cycle is much faster than our background switching rate, we can apply a moving average over 100 time steps for both the non-Cu and Cu chip to improve the signal-to-noise ratio for these time traces. We then perform a hidden Markov model analysis to identify the parity switches (see Methods).

As we vary the pulse length from 20 ns to 400 μs, we increase the injected energy and thus the number of pair-breaking phonons coupled to the chip. Longer injection pulses result in a higher rate of parity switches, with almost all of the parity switches synced with the phonon

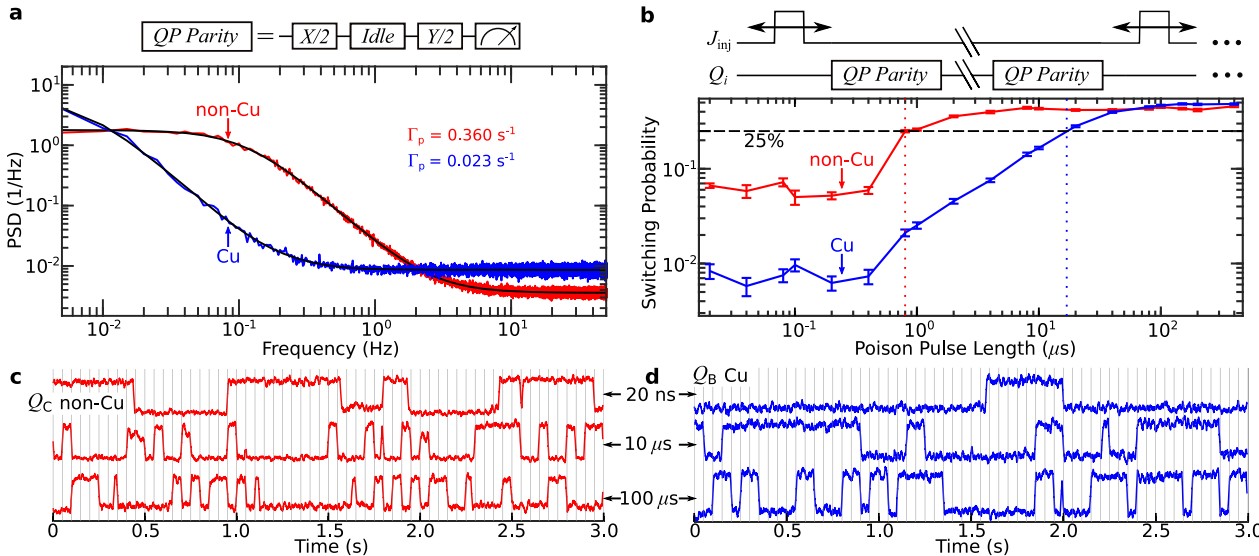

**Fig. 3 | Measurement of QP parity switching. a** Power spectral density of QP parity switching with no injection pulses for $Q_A$ on non-Cu (red) and Cu (blue) chips. **b** Measured probability of parity switch per injection pulse vs. pulse duration for non-Cu and Cu chips; dotted/dashed lines indicate pulse lengths corresponding to 25% switching probability. Error bars computed from standard Poisson counting

errors (see Supplementary Note 9). Pulse sequence for QP parity measurements without/with controlled phonon injection shown above plots in **a/b**. Segment of time series of QP parity for different injection pulse durations for (**c**) non-Cu chip, (**d**) Cu chip; vertical lines indicate timing of injection pulses.

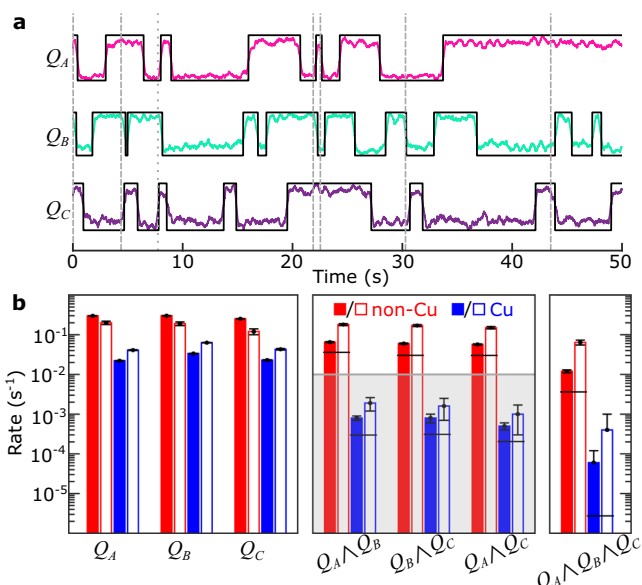

**Fig. 4 | Correlated parity switches from background radiation. a** Typical time trace of simultaneous QP parity measurements on non-Cu chip with no injection pulses. Black lines correspond to extracted digital signal from parity switches; vertical dashed (dotted) lines indicate double (triple) coincidences. **b** Observed parity switching rates (solid bars) and extracted poisoning event rates (open bars) for non-Cu (red) and Cu (blue) chips in the absence of controlled phonon injection. Expected random background coincidence rates are plotted as horizontal black lines. Error bars computed from Poisson counting errors (see Supplementary Notes 10 and 11). Fault-tolerant level for two-fold correlated errors, as described in the text, is indicated by gray-shaded region.

injection pulses [Fig. 3c, d]. In Fig. 3b we plot the ratio of the measured switching rate to the rate of phonon injection as a function of the injection pulse duration; this quantity corresponds to the probability of a measured parity switch per phonon injection pulse. For sufficiently high injected energy, we expect that pair-breaking phonons will randomize the parity on every qubit island on a timescale much shorter than our 100-μs sampling period. Because we observe a change of QP parity only for an odd number of switches, we expect our measured parity switching rate to saturate at half the injection rate for long injection pulses. As expected, our measured probabilities saturate around 0.5; however, the Cu chip requires roughly 20 times the injection energy to achieve the same level of poisoning as the non-Cu chip. If we assume that each injection pulse generates a number of pair-breaking phonons in the Si that is proportional to the pulse duration, this indicates that the Cu islands on the back side of the Cu chip downconvert 95% of the injected phonons.

**Multi-qubit correlated parity switching**

We next perform simultaneous parity measurement of all three qubits on each chip and we analyze the resulting time series to identify coincidences (see Supplementary Notes 9 and 10). We first apply this approach to the measurements with periodic QP poisoning from controlled phonon injection. For sufficiently long injection pulses, we expect the probability of double coincidences to saturate at $(1/2)^2$ and for triple coincidences to saturate at $(1/2)^3$; this is indeed what we observe (see Supplementary Note 8).

After confirming that our analysis successfully detects coincidences induced by controlled injection, we next apply this same approach to measurement of correlated poisoning induced by environmental radiation. Since the background poisoning rates are quite low for these devices, we reduce the experimental duty cycle to 10 ms and acquire simultaneous parity data over several days to build up

sufficient statistics to detect coincidences. Figure 4 presents our results from these measurements. For all three qubits on each chip, the single-qubit parity switching rate is consistent with our previously measured $\Gamma_P$ values. Based on the observed parity switching rates, we calculate the expected random double- and triple-coincidence rates (see Methods). For the non-Cu chip, the expected random rates for double (triple) coincidences are less than the observed coincidence rates by nearly a factor of 2 (3), indicating the presence of significant correlated switching.

Based on the analysis in refs. 6, 7, in the absence of mitigation, we expect correlated poisoning events to be dominated by γ impacts that broadcast high-energy phonons throughout the entire chip. For a given impact rate $R_γ$, we thus expect a rate of individual qubit parity switches $R_γ/2$, a rate of two-fold coincidences $R_γ/4$, and a rate of three-fold coincidences $R_γ/8$. We solve a system of equations for the observed coincidence probabilities to obtain the actual exclusive rates for single-, double-, and triple-qubit poisoning events (see Supplementary Note 11). If there were no other poisoning mechanisms and no phonon loss in the chip so that each γ impact poisoned all qubits with 100% probability, we would expect an extracted event rate for $Q_A \wedge Q_B \wedge Q_C$ equal to $R_γ$, while all single- and two-fold poisoning rates would be zero, since all particle impacts are expected to couple to all qubits via high-energy phonons.

Figure 4b presents the observed parity rates and extracted poisoning event rates for both chips. For the non-Cu chip, the extracted three-qubit event rate is high [0.064(9) s⁻¹], indicating the presence of significant correlated poisoning between widely separated qubits. However, the event rates for double- and single-qubit poisoning are also significant; we note that for the three qubit pairs with different physical separations, there is no clear dependence of two-fold correlated poisoning on the distance between qubits. For any practical implementation, there will always be some degree of phonon loss, for example, from the anchoring points where the chip is attached to the sample enclosure or through wirebonds, so that even in the absence of phonon downconversion structures, not all qubits are poisoned by each γ impact. In this case, $R_γ$ could be estimated as 1.1 s⁻¹, the sum of all the poisoning rates for the non-Cu chip in Fig. 4b. For the Cu chip, all of the extracted correlated event rates are two orders of magnitude lower than for the non-Cu chip; the sum of all poisoning rates on the Cu chip is 0.15 s⁻¹, which is dominated by the single-qubit poisoning rates. This indicates that the Cu reservoirs greatly reduce the footprint of the phonon burst following a high-energy particle impact.

## Discussion

We have separately performed repeated charge tomography for one qubit on each chip and observed a rate of large offset charge jumps of 0.0012(1) [0.0011(1)] s⁻¹ for the non-Cu [Cu] chip. Following the detailed modeling and analysis in ref. 6, we estimate the rate of γ impacts on our chips to be 0.083(8) s⁻¹ (see Supplementary Note 13). Thus, the higher total poisoning rate, particularly on the non-Cu chip, compared to the estimated $R_γ$ from the offset-charge measurements suggests the presence of additional phonon-mediated poisoning mechanisms in our device. THz photons above $2\Delta_{Nb}$ from warmer portions of the cryostat could break pairs in the Nb ground plane and couple phonons into the substrate from recombination, thus poisoning nearby qubits, but without the chip-wide burst of phonons from a high-energy γ impact. Additionally, the cryogenic dark matter detection community has observed heat-only events that are attributed to mechanical cracking processes in the device and sample enclosure that release stresses, typically at the attachment points[26,27]; recent work reported heat-only events in superconducting transition edge sensors on a Si chip attached to a sample holder with GE varnish[28]. Such events can produce large bursts of phonons that are detected by the sensors in these experiments, but with no accompanying charge signal. The dynamics of such heat-only events will depend on the details of the

device and enclosure design, but could potentially occur in our qubit chip and sample enclosure and serve as another phonon-mediated QP poisoning mechanism. The overall reduced poisoning rates on the Cu chip indicate that the normal metal structures reduce phonon-mediated poisoning from other mechanisms in our system, such as THz photons or heat-only events, as well.

Excess QPs cause both enhanced parity switching [Fig. 3b] and reduced $T_1$ (Fig. 2), thus resulting in enhanced bit-flip errors[8]. Thus, our demonstrated suppression of correlated QP poisoning from phonon downconversion provides a strategy for reducing correlated errors in large qubit arrays. For robust error detection, we require single-qubit errors below the $10^{-4}$ level, which will correspond to random error coincidences between pairs of qubits at the $10^{-8}$ level. Thus, any correlated two-qubit errors must be below the $10^{-8}$ level[6]. If we assume a surface code duty cycle of 1 MHz and take our largest extracted two-fold poisoning event rate of 0.002 s$^{-1}$, we find a two-fold error probability of $2 \times 10^{-9}$. Thus, our initial attempt at correlated error suppression by phonon downconversion already yields a correlated error rate below the threshold necessary for fault-tolerant operation. Further optimization, including an investigation of the dependence of the downconversion efficiency on metal film thickness and composition, and incorporation of additional mitigation strategies should guarantee the robust operation of error-corrected quantum processors in the presence of low-level pair-breaking radiation.

## Methods
### Device fabrication
Both the non-Cu and Cu chips are fabricated from high-resistivity (>10 kΩ·cm) Si wafers. For the Cu chip, the wafer is double-side polished to allow for fabrication of the Cu reservoirs. Deep-UV photolithography is used to pattern the ground plane, feedline, readout resonators, qubit islands, charge-bias lines, and injector junction pads, followed by reactive ion etching of the Nb film. After the base-layer processing, the Cu reservoir fabrication on the wafer with the Cu chip is started by first preparing a protective resist layer on the device surface, then evaporating a seed layer of Ti and Cu on the opposite side. We electroplate a 10-μm thick film of Cu on top of the seed layer; we pattern the Cu reservoirs by dicing (200 μm)$^2$ islands with partial dicing saw cuts that extend 20 μm into the back surface of the Si (see Supplementary Note 1). The qubit and injector junctions on both chips are Al-AlO$_x$-Al junctions made by double-angle evaporation, producing qubit frequencies in the range of 4.7–5.3 GHz (Supplementary Table 1).

### Measurement setup
Measurements on both the non-Cu and Cu chips are performed on the same dilution refrigerator cooldown running at a temperature below 15 mK. The Al sample boxes for both chips are mounted on the same cold-finger inside a single Cryoperm magnetic shield. A Radiall relay switch on the output lines of the two devices allows us to switch between measurements of one chip or the other. Supplementary Fig. 2 details the configuration of cabling, attenuation, filtering, and shielding inside the cryostat, as well as the room-temperature electronics hardware for control and readout. The inner surfaces of the Cryoperm magnetic shield and the mixing chamber shield were both coated with an infrared-absorbent layer[29]. For the charge-biasing of the qubits, wiring limitations on our dilution refrigerator prevented us from connecting to all of the bias traces on the chips. For the non-Cu chip, charge-bias lines are connected to $Q_B$ and $Q_C$; for the Cu chip, there is only a bias connection to $Q_A$.

### Relaxation and injection measurements
Phonon injection experiments are done by pulsing the bias on one of the Josephson junctions near the edge of each chip [Fig. 1 and Supplementary Fig. 1c] followed by a measurement of qubit $T_1$, from which we compute $\Delta\Gamma_1$ (see Supplementary Note 5). In addition to analyzing the response of $\Delta\Gamma_1$ with bias-pulse amplitude and delay between the pulse and $T_1$ measurement, we compute the change in reduced QP density, $\Delta x_{qp} = \pi\Delta\Gamma_1/\sqrt{2\Delta_{Al}\omega_{01}/\hbar}$[19], where $\omega_{01}$ is the qubit transition frequency.

### Single-qubit parity measurements
The Ramsey pulse sequence that we use for mapping QP parity onto qubit 1-state occupation is as follows: apply a $X/2$ pulse, idle for a time corresponding to a quarter of a qubit precession period, then apply a $Y/2$ pulse, followed by a qubit measurement[14,15,23]. If the offset charge is at maximal charge dispersion, the final $Y/2$ pulse projects the qubit to either the 0 or 1 state, dependent on the QP parity. In order to have an uninterrupted measurement sequence, active stabilization of the offset charge is not performed. The power spectral densities of the QP parity switching are computed from records of 20,000 single shots of the parity-mapping pulse sequence measured at a repetition period of 10 ms. We apply a simple thresholding scheme based on the 0/1 readout calibration for each qubit to generate a digital time trace of the QP parity. We then compute the PSD of this digital signal and average 20–160 of these curves to obtain Fig. 3a for $Q_A$ on each chip and Supplementary Fig. 7 for all three qubits on both chips. Since the offset charge is not actively stabilized, when the offset charge randomly jumps near the degeneracy point, the parity readout fidelity vanishes. This results in an enhancement of the white noise floor, but the characteristic QP parity switching rate $\Gamma_p$ can still be extracted (see Supplementary Note 7).

In addition to the PSD measurements, we also study single-qubit QP parity switching with periodic phonon injection [Fig. 3b–d]. Here, we simultaneously produce phonons by pulsing the injector junction to an amplitude of 1 mV at a frequency of 20 Hz while recording single shots of the QP parity-mapping pulse sequence at a repetition period of 100 μs for a duration of 400 s. As with the PSD measurements, we do not actively stabilize offset charge, and thus the offset charge will occasionally jump randomly to near degeneracy where the QP parity cannot be discriminated. In order to process the data, we apply a moving average of 100 time steps to the QP parity traces. The portions of the averaged parity traces where the peak-to-peak amplitude is below a threshold determined by the 0/1 readout calibration levels are masked off and not analyzed further. Next, a hidden Markov model (HMM) is used to identify the QP parity. We assign a probability for the parity signal to have an odd- or even-parity state based on Gaussian fits to the 0/1 readout calibration measurements for each qubit. The probability for the states to transition is set by the $\Gamma_p$ from the corresponding PSD for each qubit. We then use the Viterbi algorithm to fit a digital signal to the averaged QP parity data (see Supplementary Note 9).

### Multi-qubit parity measurements
For measurements of multi-qubit QP parity switching due to background radioactivity, we perform the QP parity-mapping pulse sequence for all three qubits on a chip simultaneously at a repetition period of 10 ms. We use the previously described HMM to identify QP parity switching from the time trace for each qubit using a moving average of 40 points. This results in the averaged QP parity switching events having a sloped step, with the width of each parity switch approximately equal to the number of points used in the moving average. Following the HMM extraction of digital parity switching traces, we identify a coincidence switching event between qubits to occur when the digital time traces switch within a window of 40 data points (see Supplementary Fig. 10). The coincident events are indexed with the relevant qubits involved in the switching event $Q_A \wedge Q_B$, $Q_B \wedge Q_C$, $Q_A \wedge Q_C$, or $Q_A \wedge Q_B \wedge Q_C$. We restrict each switch of a given qubit to participate in only one event per coincidence type. For example, a $Q_B$ switch cannot be used for two $Q_A \wedge Q_B$ coincidences, but could be used for a $Q_A \wedge Q_B$ coincidence and a $Q_B \wedge Q_C$ coincidence.

The switching rate for each type of coincidence event $r_i$, where $i = AB, BC, AC, ABC$, is given by $N_i/\tau_i$, where $N_i$ is the total number of events and $\tau_i$ is the total duration of unmasked data for event type $i$. Note that double coincidences between qubits $j$ and $k$ are only counted during the period when both qubits are unmasked; similarly, triple coincidences require that all three qubits are unmasked. The uncertainty in $r_i$ comes from the standard Poisson counting errors $N_i^{1/2}/\tau_i$ (see Supplementary Note 10).

### Extraction of correlated poisoning rates

For a set of observed single-qubit parity switching rates with a particular non-zero window $\Delta t$ for identifying coincidences, one would expect a rate of random uncorrelated coincidence switching given by the product of the probabilities for observing a parity switch for each constituent qubit during the interval $\Delta t$. These expected background coincidence parity switching rates are listed in Supplementary Table 2. The error bars for these random background coincidence rates were computed by summing the fractional uncertainty for each observed rate in quadrature. While the quantities we measure in our simultaneous parity measurements are the observed parity switching rates, we would like to compute the actual poisoning event rates $r_i$ for each qubit, or group of qubits, exclusively. For example, a single poisoning event that couples to both $Q_A$ and $Q_B$ will contribute to $r_{AB}$ but will not contribute to $r_A$ or $r_B$. Based on these criteria, we use the observed parity switching rates $r_i^{\mathrm{obs}}$ to compute the probability for observing each type of parity switching event in a window interval $\Delta t$ as $p_i^{\mathrm{obs}} = r_i^{\mathrm{obs}} \Delta t$. We then derive expressions for the probability of observing each type of parity switching event in terms of the actual probability for each type of poisoning event, as listed in Supplementary Eq. (2). We numerically solve the system of equations to obtain the actual poisoning probabilities $p_i$ and then calculate the extracted poisoning rates $r_i = p_i/\Delta t$ reported in Fig. 4b and Supplementary Table 2. The error bars on each actual poisoning probability are calculated by numerically computing the derivative with respect to each of the observed switching probabilities, then multiplying by the corresponding Poisson error bar for the observed switching probability and summing these together in quadrature (see Supplementary Note 11).

## Data availability

Data used in this work is available on https://doi.org/10.5281/zenodo.7249678. Supplementary data is available upon reasonable request.

## Code availability

Code used in this work is available upon reasonable request.

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

## Acknowledgements

This work is supported by the U.S. Government under ARO grant W911NF-18-1-0106. Fabrication was performed in part at the Cornell NanoScale Facility, a member of the National Nanotechnology

Coordinated Infrastructure (NNCI), which is supported by the National Science Foundation (Grant NNCI-2025233).

## Author contributions
V.I. and J.K. took and analyzed the data. V.I. and A.B. designed and fabricated the devices. J.K., C.P.L., and E.Y. developed code for analyzing parity switching data. C.H.L., S.P., and R.M. performed the CST Microwave simulations. J.K., V.I., and A.B. helped to develop the measurement and fabrication infrastructure. B.L.T.P. designed the experiment and directed data-taking and analysis. V.I., J.K., C.P.L., E.Y., R.M., and B.L.T.P. co-wrote the manuscript.

## Competing interests
Authors B.L.T.P. and R.M. declare a competing interest in the form of a utility patent application, FABRICATION OF NORMAL CONDUCTING OR LOW-GAP ISLANDS FOR DOWNCONVERSION OF PAIR-BREAKING PHONONS IN SUPERCONDUCTING QUANTUM CIRCUITS (application number 17/469,380), filed by the Wisconsin Alumni Research Foundation on September 8, 2021. B.L.T.P. and R.M. are the sole inventors on the application, which is currently pending. The invention covered by the patent application concerns the technique for fabricating the normal metal structures for phonon downconversion. This article describes experimental work by our two research groups demonstrating the effectiveness of this technique for phonon downconversion. The remaining authors declare no competing interests.
