## [Peer Review File · Nature Communications]

REVIEWERS' COMMENTS

Reviewer #1 (Remarks to the Author):

I would like to thank the authors for the very detailed reply and care taken to clarify the manuscript. Following my earlier review, I also support publication in Nature Communications. I only have a minor suggestion based on the response letter.

For the adhesive: please also mention the use of GE Varnish in the paper, not only in the reply. There was a recent work (arXiv:2208.02790) where Varnish was suggested as a potential release of energy bursts, which nicely connects with the discussion of the authors in p5-6 of the response letter, and also with a comment of the authors later in the response letter.

I do not suggest a detailed analysis, since this work appeared after the authors' manuscript, but it shows the importance that we report in real detail how chips are mounted.

Reviewer #2 (Remarks to the Author):

I did review the previous version of the manuscript by Iaia et al and I have found the revised manuscript satisfactory with the recent modifications/additions to address the comments I and other reviewers made. I believe the manuscript is of interest to those focused on not only superconducting quantum computing but also fields of high energy and condensed matter physics. I recommend this manuscript to be published in Nature Communications.

Reviewer #3 (Remarks to the Author):

I reviewed the newly submitted manuscript, and my original review of this work, for Nature Physics, still remains. The authors have addressed the minor concerns that I had raised, and I recommend publishing.

This manuscript presents a technique for reducing the impact of spurious quasiparticle generation in superconducting qubits that results in correlated errors, by placing a normal metal layer (Cu) on the back of the silicon chip to provide a means for energy down-conversion of high-energy phonons. The authors present extensive measurement results and statistical analysis for qubit error rates in chips with and without this mitigation and demonstrate that correlated error rates can be substantially reduced.

Although not entirely original, the conclusions are very relevant to several fields, in particular, quantum computing, and MKID detectors for astrophysical telescopes (although this area is not mentioned at all). The idea of quasiparticle poisoning due to stray radiation, whether from radioactivity, cosmic muons, or stray high-frequency radiation, and mitigating the causes of these by placing normal-metal layers on the back of the silicon chip is not new. This concept has been studied and applied extensively by the astrophysical detectors (MKID) community (who have already implemented this in working astrophysical spectrometer instruments that employ sensitive detectors such as MKIDs). However, making hard conclusions about the nature and sources of these unwanted quasiparticles has been difficult due to the complex nature of experiments and environments, and the difficulty of isolating various effects, such as stray radiation from other sources of QP poisoning, such as nuclear radiation from the enclosure. In this sense, conducting a well-controlled experiment requires a very well-thought testbed, and careful considerations for RF and magnetic shielding, which the authors of this work seem to have performed well. In this sense, the conclusions and the careful statistical analyses and simulations for qubit parity error rates are substantial enough to be worthwhile publishing here. The manuscript provides sufficient statistical analysis for the measurement data presented and the error bars in the plots seem to be appropriately indicated.

The results of this work are also highly relevant to the astrophysics detector and instrumentation community (for example, this article: <https://www.nature.com/articles/ncomms4130> and this one: <https://www.nature.com/articles/s41550-019-0850-8>).

Overall the manuscript is excellently presented and is clear and easy to read. I don't have any major issues and my prior minor concerns have been addressed. I recommend publishing.

Authors' response:

We would like to thank the three reviewers for again evaluating our manuscript entitled “Phonon downconversion to suppress correlated errors in superconducting qubits” that we submitted to *Nature Communications*. We are pleased that all three reviewers gave a positive assessment of our work and recommend publication in *Nature Communications*. Only Reviewer #1 requested any further changes to our manuscript before publication. We have added the requested reference to the recent preprint arXiv:2208.02790, as well as a brief description of the phonon-only events observed in this work into the first paragraph of our new Discussion section. Reviewer #3 mentioned two references from the astrophysical detector community, but did not explicitly request that we cite these. While the detectors described in these references would likely benefit from the phonon downconverting structures described in our manuscript, we feel that adding citations to these works would require a new section of our conclusions and would change the focus of our results. We have thus chosen not to add citations to either of these references.